# Influence of Backing Plate Support Conditions on Armor Ceramic Protection Efficiency

**DOI:** 10.3390/ma13153427

**Published:** 2020-08-03

**Authors:** Bowen Zhang, Yangwei Wang, Shaofeng Du, Zhikun Yang, Huanwu Cheng, Qunbo Fan

**Affiliations:** 1School of Materials Science and Engineering, Institute of Technology, Beijing 100081, China; zhangbw@bit.edu.cn; 2National Key Laboratory of Science and Technology on Material under Shock and Impact, Beijing 100081, China; chenghuanwu@bit.edu.cn (H.C.); fanqunbo@bit.edu.cn (Q.F.); 3Beijing Institute of Technology Chongqing Innovation Center, Chongqing 401147, China; 4State Key Laboratory of Smart Manufacturing for Special Vehicles and Transmission System, Baotou 014000, China; dusf.yx@163.com

**Keywords:** composite ceramic armor, armor structure design, armor ceramic protection efficiency

## Abstract

The bilayer composite ceramic armor is widely used in the world, while the protection efficiency of the armor ceramic in it still confuses researchers. This study applied a numerical simulation method to produce a general equation that describes the relationship between the protection efficiency of the armor ceramic and the supporting conditions of the backing plate, thereby enhancing the current understanding of the composite ceramic armor. The results indicated that the protection efficiency of the armor ceramic can be divided into three parts: (1) the basic protection efficiency, (2) the increment efficiency caused by inertial support, and (3) the increment efficiency caused by mechanical support. The inertial support is related to the density and thickness of the backing plate, and the mechanical support is related to the elastic modulus and yield strength of the backing plate materials. The inertial support exhibited a positive correlation with the protection efficiency of the armor ceramic before it reached the *S_cr_*; after that, the protection efficiency of the armor ceramic would remain stable. In addition, the mechanical support showed a linear, positive correlation with the backing plate stress at *ε*_0_.

## 1. Introduction

Armored vehicles have been considered the core of weapon studies, given that they are the main army equipment. High mobility and high-protection capabilities are key development trends in armored vehicles, particularly during complex battlefield situations. These requirements consider the armor as one of the most important parts of the vehicle, such that many companies and studies have examined the development of lighter weight and higher protection efficiency armor. The military highly considers the use of armor ceramic material due to its lightweight and excellent protection efficiency, since its first appearance in the 1960s, and the Florence model [1] was applied for the optimization of two-component composite armor. It has been widely applied in armored vehicle production for its significant military and economic benefits.

A composite armor containing armor ceramics as the main unit is termed composite ceramic armor. The composite ceramic armor protection efficiency can be divided into two aspects, namely, the protection efficiency of the backing plate and armor ceramics [2,3]. Comprehensive consideration of these two parts renders the complete design method for the production of composite ceramic armor. In general, the backing plate material is commonly used as armor material. Its protection efficiency can be estimated using current empirical equations, such as the Water-ways Experiment Station (WES) equation [4,5], the Forrestal equations [6,7,8,9,10,11], or the National Defense Research Committee (NDRC) equation [12]. However, the protection efficiency of armor ceramics relates to a world of factors that are not totally clear. In the original Florence model, and some later research, the protection efficiency of armor ceramics is ignored [13,14,15,16], which makes the armor ceramics just a reinforced part of the backing plate. However, a number of recent research studies show that armor ceramics play the core role in the composite ceramic armor [17,18,19,20,21,22,23], and the support of the backing plate material is an important prerequisite for the protective role of armor ceramic materials [24,25,26,27,28].

Replacement of the backing plate materials or changing the backing plate structure in a composite ceramic armor changes the protection efficiency of not only the armor ceramics but also the backing plate. This complex influence hinders the experimental characterization of the influence of the backing plate support toward the armor ceramic’s protection efficiency and requires additional numerical simulation technology for research and analysis. Here, the finite element method (FEM) is a widely used numerical simulation method [29,30] for its high dependability in dynamic analyses, which can help deal with this problem.

This study applied a reliable numerical simulation technology to simulate the penetration process of composite ceramic armor. The influence of the backing plate support toward the armor ceramic’s protection efficiency was examined following adjustments applied on the elastic modulus, density, yield strength, and thickness of the backing plate in the composite ceramic armor.

## 2. Numerical Simulation Method

Early research applied a numerical simulation technology on the penetration process of composite ceramic armor, specifically using LS-DYNA software (ANSYS, v14.5, Canonsburg, PA, USA) with the FEM to characterize and compare the deviation rate with the actual experimental test results. In Gao’s work [31,32], the numerical simulation method was used to study the phenomenon of dwell when a bullet strikes the armor ceramic. The location of the bullet tail in a simulation result is almost the same as that in a high-speed X-ray camera photo and high-speed camera photo, which are shown in Figure 1.

The Lagrangian element is an erosion algorithm used to remove the elements experiencing large distortions [33]. In our study, for simulating large deformation of the large-scale projectile and saving cost, Lagrangian elements are adopted for all materials. The Johnson–Holmquist 2 (JH2) constitutive model, shown in Table 1, was used for the SiC ceramic materials in consideration of tensile and compressive strength and failure behavior of ceramics, and the Johnson–Cook (JC) constitutive model was used for the backing plate materials. The solid 164 element model was used, since it contained 48 Degree of Freedom (DOF), which is much more than the 24 DOF of a normal 8-node element. To be as close to the experiment as possible, the boundary conditions were only set at the back edge of the backing plate. The global contact is set to the single surface contact, which can describe the processing of bullet penetration accurately. This numerical simulation technology was used to simulate the penetration experiments on the composite ceramic armor structures that contained a silicon carbide face plate and steel backing plate at different support conditions.

Several experiments have been done to support the dependability of the numerical simulation technology; the experimental device is shown in Figure 2. The bullet is a 7.62 mm × 54 R style Armor Piercing Incendiary (API) (The research institute 53 of CNGC, Jinan, China), the velocity of the bullet is 808 ± 8 m/s, and the initial kinetic energy is 1649 J. The material of the ceramic plate is silicon carbide (Fuzhou Qiyue ceramic micro powder Co. LTD, Fuzhou, China), the material of the backing plate is 685 steel (Beijing Iron and Steel Research Institute, Beijing, China), 45# steel (Beijing Iron and Steel Research Institute, Beijing, China), or TC4 titanium alloy (Baoti Group Co. LTD, Baoji, China), and the material of the witness plate is Rolled Homogeneous Armor (RHA) or 6061 aluminum alloy (Aluminum Corporation of China, Beijing, China).

The thickness of the ceramic plate is 3 mm or 6 mm, the thickness of the backing plate ranged from 2 to 6 mm to make sure the simulation method is fit to all situations, and the thickness of the witness plate is 30 mm, which can be regarded as semi-infinite thickness. For the RHA witness plate, the residual penetration depth is measured, which is shown as P_B_ in Figure 3. Meanwhile, the volume of penetration is measured for a 2024 aluminum alloy witness plate, which is shown as Γ_B_ in Figure 3. Table 2 presents the penetration depth comparison results for the RHA witness plate, wherein the deviations between the numerical simulation and experimental test results were very small, indicating the high reliability of the numerical simulation method. Moreover, Table 3 presents the volume of penetration on a 2024 aluminum alloy witness plate, in which the numerical simulation results are close to the average experimental test results, since the errors of volume are all less than 15%.

## 3. Effect of the Backing Plate Material Parameters on the Energy Dissipation of Armor Ceramics under a Semi-Infinite Thickness

The backing plate exhibited negligible plastic deformation during penetration before the ceramic material was completely damaged. Therefore, the present study mainly considered the elastic deformation phase for its great influence on the protection efficiency of the armor ceramic material. Other mechanical properties of the backing plate material at the elastic deformation phase can be calculated from the properties above and do not require separate listings.

Figure 4 shows the kinetic energy of the penetrator and the internal energy of both the ceramic and the backing plate during the penetration. The internal energy of the ceramic and backing plate showed the interaction between the armor and the penetrator. It can be seen in Figure 4 that the internal energy of ceramic is very low after the penetrator touches the backing plate. So, the protection efficiency of the armor ceramic material was characterized by the decrease of the penetrator kinetic energy as it touched the backing plate, as it shows in Figure 4. The influence of the material parameters of *E*, *σ_y_*, and *ρ* on the ceramic panel on the semi-infinitely thick backing plate was studied under a numerical simulation. A set of 4-mm SiC ceramics were used as the ceramic surface layer, while the high-strength armor steel was selected as the reference parameter, which are listed in Table 4.

The value range of each parameter covered the known parameter range of the solid materials, or it was even much higher than the known solid materials, which can effectively guide the application and guide the study of armor materials at same time. The numerical simulation results based on this value range are shown in Figure 5. The yield strength and elastic modulus of the semi-infinitely thick backing plate greatly influenced the protection efficiency of the armor ceramic. In comparison, a lower density indicated a different influence on the protection efficiency of the armor ceramic as compared to the yield strength and elastic modulus influence.

The semi-infinite thickness of the backing plate concealed the influence of the density parameters, indicating that the density and thickness of the backing plate jointly determined a part of the backing plate support. This specific part of the support was unrelated to the mechanical properties and was related to the basic physical properties of the backing plate material, such that this study identified it under the term inertial support. The yield strength and elastic modulus still significantly affected the protection efficiency of the armor ceramic of the semi-infinitely thick backing plate, indicating a different influence from the inertial support. The present study identified the yield strength and elastic modulus as mechanical support given that they are both mechanical performance parameters.

## 4. Influence of the Inertial Support on the Protection Efficiency of the Armor Ceramic

Inertial support describes the supporting capacity of the material, which is related to the density and thickness of the backing plate. Therefore, the protection efficiency of armor ceramic was examined under a limited yield strength and elastic modulus using the parameters in Table 5, which are similar to the conditions of the semi-infinitely thick backing plate. In this part of the study, 4-mm SiC ceramics were still used as the ceramic surface layer, and high-strength armor steel parameters were used as the original parameters of the backing plate.

### 4.1. Backing Plate Thickness and Protection Efficiency of the Armor Ceramic

Figure 6 presents the protection efficiency of the armor ceramic and backing plate thickness under high-strength armor steel support. The armor ceramic without a backing plate exhibited a support protection efficiency of 378 J, which is the basic protection efficiency of the armor ceramic. Other defense efficiencies increased on this basis.

The protection efficiency of the armor ceramic backing plate thickness increased linearly from the basic protection efficiency to a certain limit value and subsequently stabilized, as summarized in Equation (1). Here, *E_c_* is the protection efficiency of the armor ceramic, *E*_0_ is the basic protection efficiency of the armor ceramic, *δ_b_* is the thickness of the backing plate, *k_c_*_1_ is a support coefficient without specific physical meaning, *E_cmax_* is the maximum protection efficiency of the armor ceramic, and *δ_bcr_* is the minimum backing plate thickness with which the protection efficiency of the armor ceramic can reach *E_cmax_*.
(1)$$E_{c}=\{\begin{array}{c} E_{0}+k_{c1}\cdot \delta_{b}\left(\delta_{b}⩽\delta_{bcr}\right)\\ E_{cmax}\left(\delta_{b}>\delta_{bcr}\right) \end{array}$$

### 4.2. Backing Plate Density and Protection Efficiency of the Armor Ceramic

Figure 7 presents the protection efficiency of the armor ceramic and the density of the backing plate with a 2-mm backing plate. The protection efficiency was very close to the basic protection efficiency when the density of the backing plate was very low and exhibited a sharp increase with the backing plate density and subsequently showed slow growth. The observed trend was exactly similar to an exponential function with an exponent of less than one.

The armor ceramic exhibited increments in its protection efficiency, with the exception of the basic protection efficiency. Double logarithmic coordinates were drawn to determine the correct equation to describe the protection efficiency increment of the armor ceramic and the density of the backing plate (Figure 8). The protection efficiency increment of the armor ceramic logarithmic increased with the backing plate density.

The equation in Figure 8 was fitted by the least squares method to generate Equation (2), in which *E_cplus_* is the increment of protection efficiency of armor ceramic, and *ρ* is the density of the backing plate.
(2)$$lgE_{c}=1.1+0.35lg\rho$$

The R-squared value of this fitting equation was greater than 0.9, further indicating that the simulation result followed linear growth. The armor ceramic protection efficiency equation for the backing plate density presented an exponent of 0.35, which was approximately equal to 1/3 of the engineering calculation. Therefore, the backing plate density and protection efficiency of the armor ceramic can be summarized shown in as Equation (3), in which *E_c_* is the protection efficiency of the armor ceramic, *E*_0_ is the basic protection efficiency of the armor ceramic, *k_c_*_2_ is a support coefficient without specific physical meaning, *ρ* is the density of the backing plate, *E_cmax_* is the maximum protection efficiency of the armor ceramic, and *ρ_cr_* is the minimum backing plate density with which the protection efficiency of the armor ceramic can reach *E_cmax_*.
(3)$$E_{c}=\{\begin{array}{c} E_{0}+k_{c2}\cdot \rho^{\frac{1}{3}}\left(\rho ⩽\rho_{cr}\right)\\ E_{c}\left(\rho >\rho_{cr}\right) \end{array}$$

### 4.3. Backing Plate Inertial Support and Protection Efficiency of the Armor Ceramic

The influences of the density and back plate thickness on the protection efficiency of the armor ceramic can be combined into one equation, as shown in Equations (4) and (5), in which *E_c_* is the protection efficiency of the armor ceramic, *E*_0_ is the basic protection efficiency of the armor ceramic, *k_m_* is a support coefficient without specific physical meaning, *ρ* is the density of the backing plate, *E_cmax_* is the maximum protection efficiency of the armor ceramic, and *S_cr_* is the minimum backing plate inertial support with which the protection efficiency of the armor ceramic can reach *E_cmax_*.
(4)$$E_{c}=\{\begin{array}{c} E_{0}+k_{m}\cdot S^{\frac{1}{3}}\left(S⩽S_{cr}\right)\\ E_{c}\left(S>S_{cr}\right) \end{array}$$
in which,
(5)$$\;S=\delta_{b}\cdot \rho^{\frac{1}{3}}$$

Figure 9 presents the backing plate inertial support and protection efficiency of the armor ceramic following the derivations of the above equations. The protection efficiency of the armor ceramic was determined by the values of $E_{0}$, $S_{cr}$, and $k_{m}$ in the equation.

*E*_0_, the basic protection efficiency of the armor ceramic, was completely dependent upon the mechanical properties of the armor ceramic, the thickness of the armor ceramic, and the characteristics of the projectile. This parameter exhibited minimal dependence with the structural design.

*S_cr_* is the criterion for the inertial support to the armor ceramic to reach its maximum. A backing plate inertial support that is higher than *S_cr_* indicates that an increase in inertial support will exhibit no additional physical gain on the protection efficiency of the armor ceramic. The value of *S_cr_* is limited by the basic mechanical properties of the armor ceramic as well as the characteristics of the projectile. *S_cr_* describes the theoretical optimal thickness of the backing plate according to its structural design. The *S_cr_* of the 6-mm and 10-mm SiC ceramics was obtained by penetrating the 12.7 mm API, to which the results are shown in Table 6.

Projectiles with similar shapes, material, and velocity exhibited minimal changes in their respective *S_cr_* per unit of ceramic thickness. In other words, the *S_cr_* approximately conformed to the similar law at a certain condition. Therefore, in the engineering calculation, the *S_cr_* per unit ceramic thickness can be regarded as a constant value for the approximate calculations.

*k_m_* is the coefficient of the inertial support, which is affected by the mechanical properties of the backing plate, the mechanical properties of the armor ceramic, and the characteristics of the projectile. Therefore, the *k_m_* of samples with the same backing plate material, projectile characteristics, and armor ceramic can be regarded as a constant value for the approximate calculations, similarly to what was described above for *S_cr_*.

## 5. Influence of the Mechanical Support on Protection Efficiency of the Armor Ceramic

The mechanical support describes the influence of the backing plate mechanical properties toward the armor ceramic protection efficiency, which can be related to the yield strength and elastic modulus of the backing plate material. Therefore, the protection efficiency of the armor ceramic was examined against changes in the backing plate mechanical properties under a limited backing plate thickness and density (Table 7), which is similar to the condition of the semi-infinitely thickness backing plate. This portion of the study still employed 4-mm SiC ceramics for the ceramic surface layer. Additionally, 2-mm high-strength armor steel parameters were used as the original parameters of the backing plate.

### 5.1. Backing Plate Elastic Modulus and Protection Efficiency of the Armor Ceramic

The influence of the elastic modulus under a logarithmic coordinate axis form against the protection efficiency of the armor ceramic was characterized with a semi-infinitely thick and a 2-mm backing plate (Figure 10).

The armor ceramic protection efficiency increased linearly with the elastic modulus of the backing plate materials under a semi-infinite thickness. However, changes in the elastic modulus significantly influenced the stress wave transmission range, given the strong dependence of the material elastic modulus on the sonic speed, thus affecting the inertial support of the backing plate. Therefore, the influence of elastic modulus on the armor ceramic protection efficiency under a semi-infinite thickness could not be employed as a perfect reference to study the effect of the mechanical support.

The protection efficiency of the armor ceramic exhibited a stable and linear increase with the elastic modulus using the 2-mm thick backing plate. However, the protection efficiency of the armor ceramic did not exhibit a low *E*_0_ even at a low backing plate elastic modulus, indicating that the ceramic layer was still supported by the backing plate under a low modulus.

### 5.2. Backing Plate Yield Strength and Protection Efficiency of the Armor Ceramic

Figure 11 presents the effects of the yield strength of the backing plate on the armor ceramic protection efficiency for the semi-infinite and 2-mm thickness backing plate. Similarly to the effect of the elastic modulus, all the trends exhibited stable, linearly increases with the yield strength, validating the observed general mechanical support trend of the backing plate.

The starting and final values as well as the lower and upper inflection points were all different between the 2-mm and semi-infinite thickness backing plates. Former research explained the difference between the starting and final values, specifically due to the contributions of the increasing inertial support. Increment differences indicate more efficient yield strength under the condition of a semi-infinitely thick backing plate. In other words, higher inertial support suggests that the efficiency of the backing plate yield strength or even mechanical support was based on the efficiency of the inertial support.

Different lower and upper inflection points indicate different mechanical support needs, such that higher inertial support requires lower mechanical support. Furthermore, the simulation experiments employed a backing plate with both an elastic modulus of 5 TPa and a yield strength of 10 GPa, thus presenting the highest mechanical support for the calculation of the 2-mm and semi-infinitely thick backing plate. The armor ceramic protection efficiency under a semi-infinitely thick backing plate reached 1074 J, which was similar to the 1089 J at 5 TPa elastic modulus and 1040 J at a yield strength of 10 GPa, indicating that the protection efficiency of the armor ceramic already reached its maximum with the semi-infinitely thick high-strength steel backing plate. Interestingly, the armor ceramic protection efficiency with the 2-mm thickness backing plate reached 850 J, which is significantly higher than 708 J at an elastic modulus of 5 TPa and 680 J at a yield strength of 10 GPa, indicating that the mechanical support is based on the complex effect of the elastic modulus and yield strength of the backing plate.

The elastic strain was determined based on the physical significance of the elastic modulus and yield strength. A certain strain limited the effect of the mechanical support (Figure 12), such that ***σ_cr_*_1_** and ***σ_cr_*_2_** were the lower and upper inflection points of the mechanical support, and ***ε*_0_** was the certain strain mentioned above. The mechanical support was only observed when the stress of the backing plate at ***ε*_0_** was above ***σ_cr_*_1_**. In addition, the mechanical support only exhibited and maintained its maximum effect when the stress of the backing plate at ***ε*_0_** was above ***σ_cr_*_2_**. No proper method is currently in place for the ***ε*_0_**, such that the value should be related to the performance of the ceramic layer and requires further examination.

According to the above observations, the relationship between the mechanical support and protection efficiency of the armor ceramic can be summarized by Equation (6), in which *E_F_* is the protection efficiency increment of the armor ceramic caused by mechanical support, *σ* is the backing plate stress at *ε*_0_, *k_F_* is a support coefficient without specific physical meaning, *E_Fmax_* is the maximum protection efficiency increment of the armor ceramic caused by mechanical support, and *σ_cr_*_1_ and *σ_cr_*_2_ are the upper and lower limit within which the change of mechanical support can influence the protection efficiency of the armor ceramic.
(6)$$E_{F}=\{\begin{array}{c} 0\left(\sigma ⩽\sigma_{cr1}\right)\\ k_{F}\cdot \left(\mathrm{lg}\left(\sigma \right)-\mathrm{lg}\left(\sigma_{cr1}\right)\right)\left(\sigma_{cr1}<\sigma ⩽\sigma_{cr2}\right)\\ E_{Fmax}\left(\sigma >\sigma_{cr2}\right) \end{array}$$

Simulation experiments without mechanical support and with maximum mechanical support were conducted at different inertial supports to further understand the effect of mechanical support, to which the results are shown in Table 8. The protection efficiency increments of the armor ceramic by the mechanical support were always 30–40% of the whole armor ceramic protection efficiency with maximum mechanical support, which can be treated as a certain value in engineering computations. In addition, Equation (7) summarizes these observations and describes the effect of mechanical support completely. Notably, *E_m_* was defined as the protection efficiency increment caused by the inertial support without mechanical support, which is different from *E_c_* mentioned above. The definition of *k_F_* was maintained when the ceramic and threat were the same, which can be taken as 2 under the study conditions. *E_F_* is the protection efficiency increment of the armor ceramic caused by mechanical support, *E*_0_ is the basic protection efficiency of the armor ceramic, *E_m_* is the protection efficiency increment of the armor ceramic caused by inertial support, *σ* is the backing plate stress at *ε*_0_, *k_F_* is a support coefficient without specific physical meaning, *E_Fmax_* is the maximum protection efficiency increment of the armor ceramic caused by mechanical support, and *σ_cr_*_1_ and *σ_cr_*_2_ are the upper and lower limit within which the change of mechanical support can influence the protection efficiency of armor ceramic, respectively.
(7)$$E_{F}=\{\begin{array}{c} 0\left(\sigma ⩽\sigma_{cr1}\right)\\ \frac{E_{0}+E_{m}}{k_{F}}\cdot \frac{\left(\mathrm{lg}\left(\sigma \right)-\mathrm{lg}\left(\sigma_{cr1}\right)\right)}{\left(\mathrm{lg}\left(\sigma_{cr2}\right)-\mathrm{lg}\left(\sigma_{cr1}\right)\right)}\left(\sigma_{cr1}<\sigma ⩽\sigma_{cr2}\right)\\ E_{Fmax}\left(\sigma >\sigma_{cr2}\right) \end{array}$$

## 6. Conclusions

(1) The protection efficiency of the armor ceramic in the composite ceramic armor can be divided into three parts, and the equation can be summarized as follows:$$E_{C}=E_{0}+E_{m}+E_{F}$$
where *E*_0_ is the basic protection efficiency, *E_m_* is the protection efficiency increment caused by inertial support, and *E_F_* is protection efficiency increment caused by mechanical support.

(2) The inertial support equation can be expressed as follows:$$E_{m}=\{\begin{array}{c} k_{m}\cdot S\left(S⩽S_{cr}\right)\\ E_{m}\left(S>S_{cr}\right) \end{array}$$
where *S* is the inertial support and can be defined as follows:$$S=\delta_{effect}\cdot \rho^{\frac{1}{3}}$$
where *δ_effect_* is the effective thickness of the backing plate, which is the smaller value between the theoretical thickness and the actual thickness. The theoretical thickness was affected by the sonic of the backing plate and the action time of the projectile.

(3) The mechanical support equation can be expressed as follows:$$E_{F}=\{\begin{array}{c} 0\left(\sigma ⩽\sigma_{cr1}\right)\\ \frac{E_{0}+E_{m}}{k_{F}}\cdot \frac{\left(\mathrm{lg}\left(\sigma \right)-\mathrm{lg}\left(\sigma_{cr1}\right)\right)}{\left(\mathrm{lg}\left(\sigma_{cr2}\right)-\mathrm{lg}\left(\sigma_{cr1}\right)\right)}\left(\sigma_{cr1}<\sigma ⩽\sigma_{cr2}\right)\\ E_{Fmax}\left(\sigma >\sigma_{cr2}\right) \end{array}$$
where *σ* is the stress when the strain of the backing plate (*ε*_0_) is affected by the thickness of the armor ceramic and the form of the projectile.

(4) Higher inertial support required lower mechanical support. As the 4-mm SiC armor ceramic was impacted by 7.62 API, the thickness of the backing plate increased from 2 mm to semi-infinite, which makes its *σ_cr_*_1_ decrease from 500 to 50 MPa and *σ_cr_*_2_ decrease from 5 to 1 GPa.

This study warrants further investigation to validate the complete theory. For example, methods used to calculate *ε*_0_, *k_m_*, and *k_F_* require changes in the thickness of ceramic or the velocity of the project. Moreover, further experimental verification is necessary to optimize the presented experimental design.

## Figures and Tables

**Figure 1 materials-13-03427-f001:**
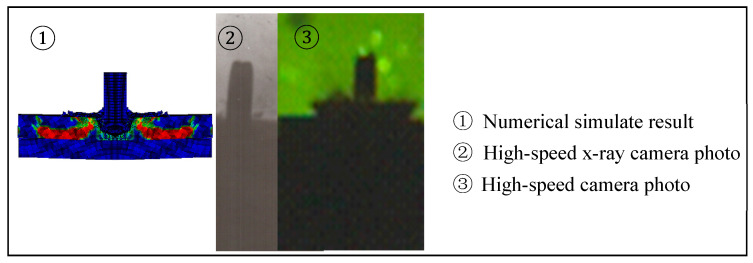
Comparison of the numerical simulation and test results of the projectile penetration [32].

**Figure 2 materials-13-03427-f002:**
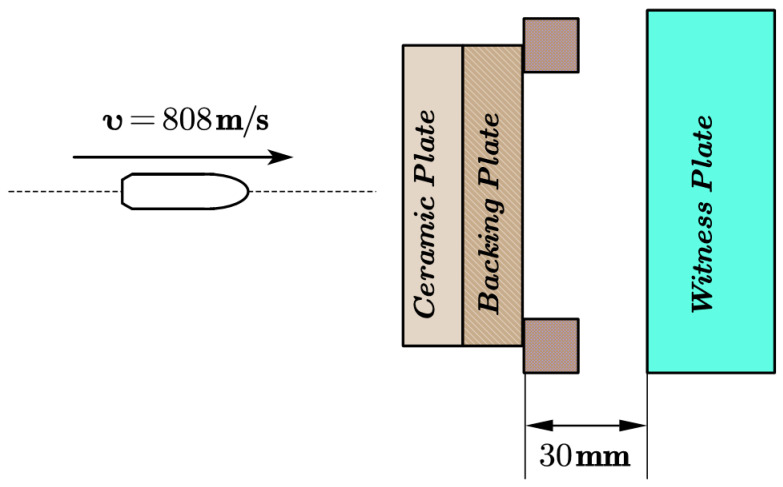
Bullet experiment device.

**Figure 3 materials-13-03427-f003:**
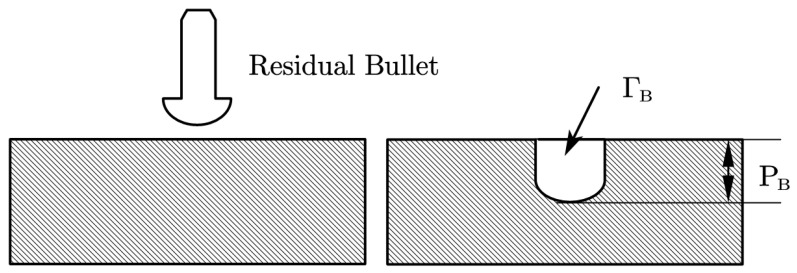
Schematic of obtaining method of experimental results.

**Figure 4 materials-13-03427-f004:**
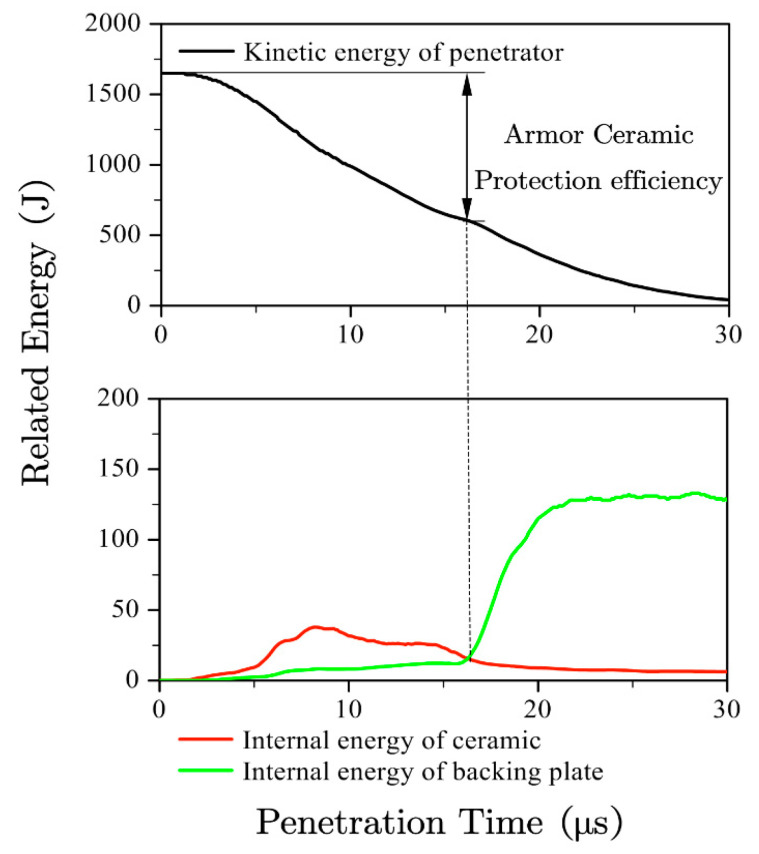
Extraction method of the protection efficiency of the ceramic layer.

**Figure 5 materials-13-03427-f005:**
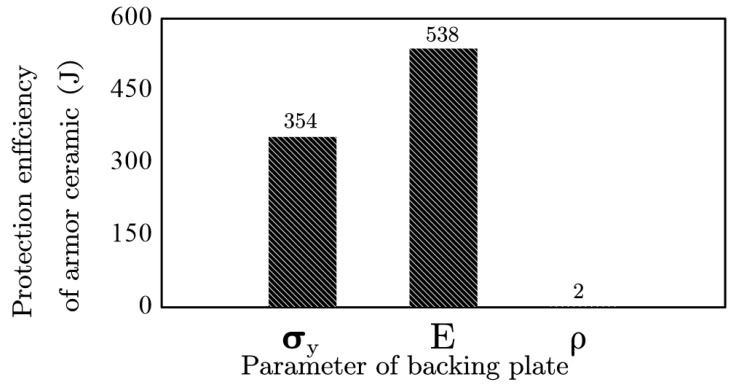
Change in the protection efficiency of the armor ceramic following the backing plate parameter changes.

**Figure 6 materials-13-03427-f006:**
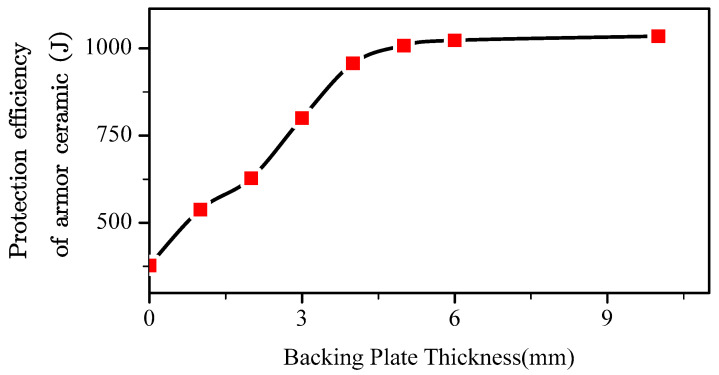
Energy dissipation of the armor ceramics at various back plate thicknesses.

**Figure 7 materials-13-03427-f007:**
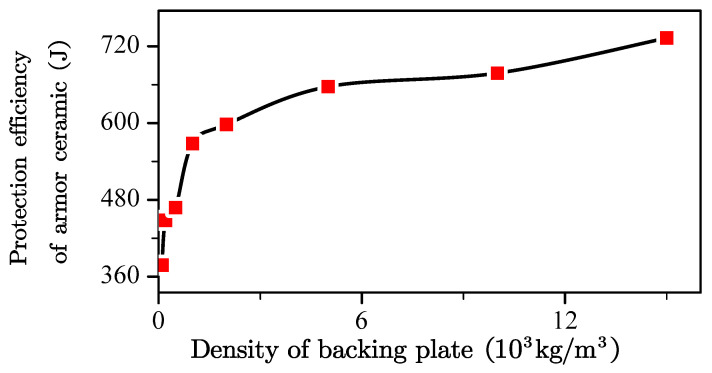
Change in the law of energy absorption and density of the ceramic layer under the 2-mm back plate.

**Figure 8 materials-13-03427-f008:**
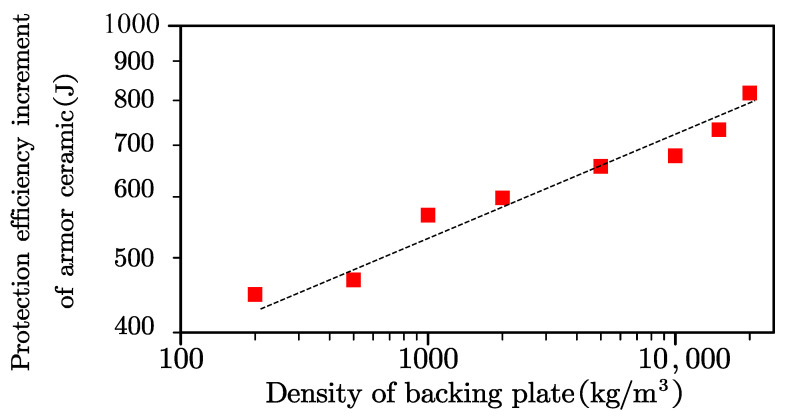
Double logarithmic coordinates describing the protection efficiency increment of the armor ceramic and density of the backing plate.

**Figure 9 materials-13-03427-f009:**
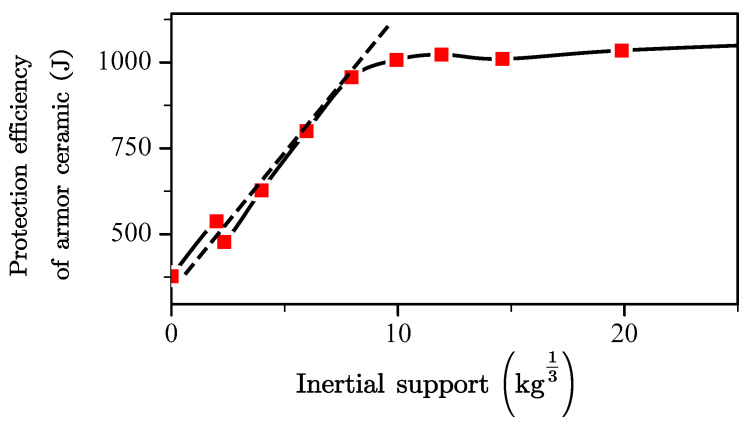
Effect of the inertia support on the protection efficiency of the armor ceramic.

**Figure 10 materials-13-03427-f010:**
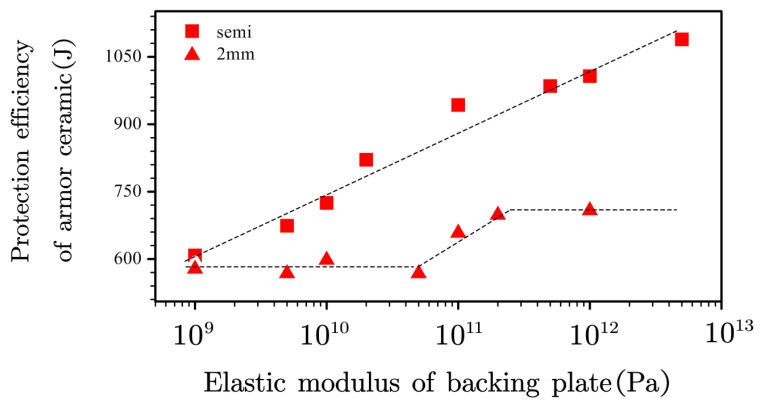
Effect of the elastic modulus of the backing plate on the protection efficiency of the armor ceramic.

**Figure 11 materials-13-03427-f011:**
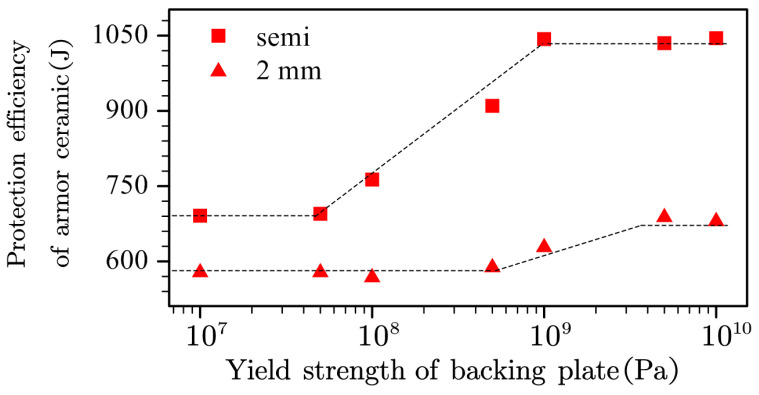
Effect of the yield strength of the backing plate on the armor ceramic protection efficiency.

**Figure 12 materials-13-03427-f012:**
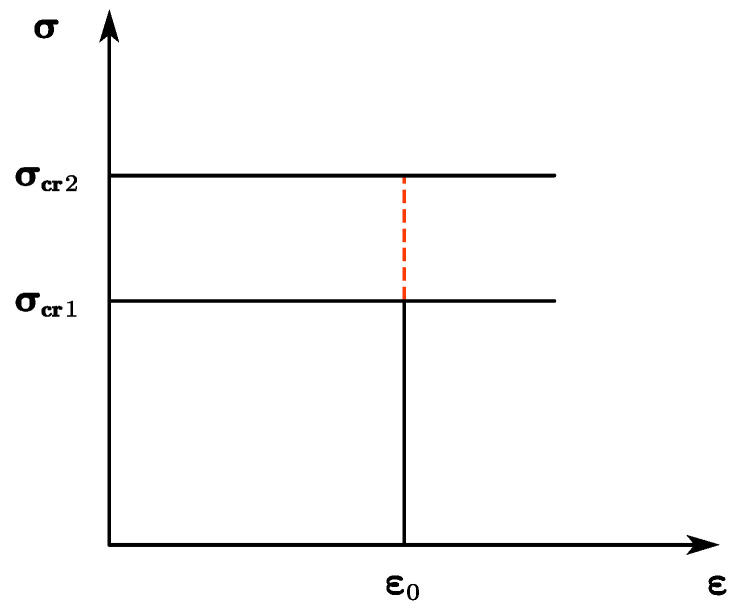
Schematic of the mechanical support.

**Table 1 materials-13-03427-t001:** Johnson–Holmquist 2 (JH2) constitutive model of SiC ceramic material.

ρ	G	A	B	C	M	*N*
3.20	1.94	0.96	0.35	0.00	1.00	0.65

**Table 2 materials-13-03427-t002:** Comparison between numerical simulation and experimental residual penetration depth on RHA witness plate under the penetration condition of 7.62 mm Armor Piercing Incendiary (API).

Ceramic Layer	Backing Plate	Numerical Simulation Results (mm)	Test Results (mm)	Error of Depth (%)
6 mm SiC	4 mm 685 steel	0.15 (on backing plate)	0.16 (on backing plate)	6.3
6 mm SiC	4 mm #45 steel	0.91	1.00	9

**Table 3 materials-13-03427-t003:** Comparison between numerical simulation and experimental volume of penetration on a 2024 aluminum alloy witness plate under the penetration condition of 7.62 mm API and a 3-mm thick ceramic layer.

Backing Plate	Numerical Simulation Results (mm^3^)	Test Results (mm^3^)	Average (mm^3^)	Error of Volume (%)
2 mm 45# steel	929	1001	877	5.9
		763		
		866		
3.5 mm 45# steel	645	581	589	9.5
		481		
		704		
5 mm 45# steel	424	462	493	14.0
		289		
		729		
2.5 mm TC4	927	1064	953	2.6
		831		
		962		
6 mm TC4	372	382	396	6.1
		410		

**Table 4 materials-13-03427-t004:** Original values and value ranges of backing plate parameters under the semi-infinitely thick conditions.

Characteristic	Modulus (GPa)	Density (10^3^ kg/m^3^)	Yield Strength (MPa)
Original value	200	7.85	1181
Value range	1–5000	0.1–25	10–10,000

**Table 5 materials-13-03427-t005:** Parameters of the different inertial supports.

Characteristic	Density (10^3^ kg/m^3^)	Thickness (mm)
Original value	7.85	2
Value range	0.1–25	0–10

**Table 6 materials-13-03427-t006:** *S_cr_* changes of the silicon carbide ceramic at different threat conditions.

Projectile	Ceramic Thickness (mm)	*S_cr_* (kg)^1/3^	*S_cr_*/Ceramic Thickness (kg/mm)^1/3^
7.62 mm API	4	7.9	1.98
12.7 mm API	6	10.8	1.80
12.7 mm API	10	18.4	1.84

**Table 7 materials-13-03427-t007:** Parameters of the different mechanical supports.

Characteristics	Modulus (GPa)	Yield Strength (MPa)
Original value	200	1181
Value range	1–5000	10–10,000

**Table 8 materials-13-03427-t008:** Energy dissipation gain of mechanical bracing under different inertial bracing conditions.

Backing Plate Thickness (mm)	Backing Plate Density (10^3^ kg/m^3^)	Inertial Support (kg^1/3^)	Basic Protection Efficiency (J)	Percentage (%)	Protection Efficiency without Mechanical Support (J)	Percentage (%)	Protection Efficiency with Maximum Mechanical Support (J)	Percentage (%)
2	2.00	2.52	378	49	488	14	770	37
2	5.00	3.42	378	44	538	19	857	37
2	7.85	3.97	378	44	578	24	850	32
2	10.00	4.31	378	44	578	23	865	33
2	15.00	4.93	378	41	638	28	919	31
2	20.00	5.43	378	40	653	29	956	32
2	25.00	5.85	378	39	676	31	960	30
5	7.85	9.94	378	37	660	28	1017	35
6	7.85	11.92	378	35	648	25	1071	40
10	7.85	19.87	378	35	682	28	1069	36
Semi	7.85	Max	378	35	691	29	1074	36

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
