# Peer review of "Influence of Backing Plate Support Conditions on Armor Ceramic Protection Efficiency"

_materials, 2020, doi:10.3390/ma13153427_

Round 1
Reviewer 1 Report
The current research reports on mechanical deformation analysis of two-layered composite plate made of armor ceramic and buckling plate using finite element (FE simulation). The composite plate is under impact load, though the numerical simulation can be extended to cover general loading conditions such as pressure, point load, etc. Three factors were taken into account for FE simulation analysis including the basic protection efficiency, the increment efficiency caused by inertial support, and the increment efficiency caused by mechanical support.
This is a worthy research which can potentially be published. However, it requires some revisions at this stage. My comments and recommendations are as follows:
1) The abstract is not informative and concise. A good abstract in a research project encompasses a summary that identifies the purpose, problem, methods, results, and conclusion of your work. Your abstract lacks a brief discussion regarding results and conclusions of your work. You have mentioned/repeated what is going to be achieved but what has been achieved is also important.
2) The authors claim to have utilized numerical (FE) study as their primary approach, yet how the FE simulation was implemented is missing from the manuscript.The authors are encouraged to discuss how the FE simulation analysis was conducted. Some issues to be considered:
-How were the loading and composite material properties introduced into the FE software?
-How were the boundary conditions prescribed to your FE model? It is of extreme importance.
-What type of element has been selected from the material library? exp: shell elements if plane stress is considered for thin plates or solid elements for thick and/or moderately thick plates.
3) Have the authors considered the material interface effect between the armor ceramic and buckling plate?
4) The literature review doesn't provide sufficient information into the core objective of this research. There are some useful researchers associated with FE analysis of composite structures as below:
"A quadratic piezoelectric multi-layer shell element for FE analysis of smart laminated composite plates induced by MFC actuators", Smart Materials and Structures, 27 095004 (2018).
"Localized failure analysis of internally pressurized laminated ellipsoidal woven GFRP composite domes: Analytical, numerical, and experimental studies", Archives of Civil and Mechanical Engineering, Vol:19, pp:1235-1250.
The authors are expected to discuss the above-mentioned papers in the introduction section of their manuscript and demonstrate their own research significance by comparing their research to them.
5) Another important issue is how did the authors compare and/or verify their FE simulation results with the High-speed x-ray camera photo? The reviewer thinks it is not practically possible as the Xray image doesn't provide anything associated with stress-strain deformation.
6) What's the reason to call the plate a buckling plate? does it mean it had been previously buckled and then assembled/connected to the armor ceramic? Please explain.
7) All FE simulation contour plots should be accompanied by a proper variable gradient such as the case in Fig.1.
8) Moderate language revision is requested to improve the manuscript's written expression.
If the above-mentioned comments are carefully addressed, the manuscript will then be acceptable for publication in my point of view.
Author Response
Dear reviewer:
Thanks so much of your advices. These advices help me a lot to improve my article.
I hope the modifications I just made is ok for you.
Please see the attachment.
And thanks for your help again, best wishes of you.

Reviewer 2 Report
Authors utilized finite element analysis methods to describe effect of backing plate properties and thickness on ballistic armor performance for ceramic hard faced vehicle armor. Model development and design are appropriate, and presented in a logical and concise fashion. Relevant governing equations are provided, and conclusions are supported by the results presented. Recommendation is for acceptance with minor corrections.
- Authors need to update plots to include axis labels.
- Discussion of backing plate simulation using with a 5 TPa elastic modulus and 10 GPa yield strength is a bit confusing. This reviewer is not aware of a material that posses these properties. A brief statement explaining why these values were chose for the simulation would be appreciated.
Author Response

(The authors gave the same response as above.)

Reviewer 3 Report
The article “Influence of backing plate support conditions on armor ceramic protection efficiency” is a parametric simulation study of the ballistic behavior of ceramic armors.
The basic idea of the paper is not bad but in overall, this paper is very confusing and not well described.
The model fig1 appears almost without description in the text, a “theory and calculation” section is definitely missing in the paper
The experimental conditions are not well described, an “experiment and method” section are also missing.
In the following, the equations and characterizations arrive without being previously defined which is very confusing.
The presentation is also very poor with fig.3,4,5,6,8,9,10 that have undefined Y, X axis.
Knowing the large amount of work to make the paper suited for a journal such as “Materials”, I recommend to reject.
Author Response
Dear reviewer:
I am so sorry that my article made you confused last time.
I know you have already rejected my article, however, I still hope for you can read this article again. I just modified it follow your advices.
It is necessary for me to recive your further advice.
Thank you very much, and allow me make an apologise again.
Please see the attachment.

Round 2
Reviewer 1 Report
The authors have revised the manuscript as per my comments. In my point of view, this version of manuscript is acceptable for publication.
Author Response
Dear reviewer:
  Thank you so much for your advice.
  Best wishes of you.
Reviewer 3 Report
The paper was improved by some minor/major revisions.
The methods and models still seem poorly introduced to me but I leave the final decision to the editor.
Author Response
Dear reviewer:
  Thank you so much to reconsider my article. To improve the article, I added some method descriptions in it. And added some explainations on the chosen of Lagrangian element.
  Hoping for your further advice.
  Thank you again, and best wishes of you.

Round 3
Reviewer 3 Report
Accept in present form